# Targeting Trypanothione Reductase, a Key Enzyme in the Redox Trypanosomatid Metabolism, to Develop New Drugs against Leishmaniasis and Trypanosomiases

**DOI:** 10.3390/molecules25081924

**Published:** 2020-04-21

**Authors:** Theo Battista, Gianni Colotti, Andrea Ilari, Annarita Fiorillo

**Affiliations:** 1Department of Biochemical Sciences, Sapienza University, P.le A.Moro 5, 00185 Rome, Italy; theo.battista@uniroma1.it; 2Institute of Molecular Biology and Pathology, Italian National Research Council, IBPM-CNR, c/o Department of Biochemical Sciences, Sapienza University, P.le A.Moro 5, 00185 Rome, Italy; gianni.colotti@cnr.it (G.C.); andrea.ilari@cnr.it (A.I.)

**Keywords:** trypanosomatid infection, structure-based drug design, trypanothione reductase, rational drug discovery, protein crystallography

## Abstract

The protozoans *Leishmania* and *Trypanosoma*, belonging to the same Trypanosomatidae family, are the causative agents of Leishmaniasis, Chagas disease, and human African trypanosomiasis. Overall, these infections affect millions of people worldwide, posing a serious health issue as well as socio-economical concern. Current treatments are inadequate, mainly due to poor efficacy, toxicity, and emerging resistance; therefore, there is an urgent need for new drugs. Among several molecular targets proposed, trypanothione reductase (TR) is of particular interest for its critical role in controlling the parasite’s redox homeostasis and several classes of active compounds that inhibit TR have been proposed so far. This review provides a comprehensive overview of TR’s structural characterization. In particular, we discuss all the structural features of TR relevant for drug discovery, with a focus on the recent advances made in the understanding of inhibitor binding. The reported cases show how, on the basis of the detailed structural information provided by the crystallographic analysis, it is possible to rationally modify molecular scaffolds to improve their properties.

## 1. Introduction

Leishmaniasis, Chagas disease, and human African trypanosomiasis (HAT), also known as sleeping sickness, are vector borne zoonosis that affect millions of people worldwide and lead to the death of about 100,000 humans per year. These diseases are caused by infection with the trypanosomatids *Leishmania* (*L.*), *Trypanosoma* (*T.*) *cruzi*, and *Trypanosoma brucei*, respectively.

Several species of *Leishmania* parasites, transmitted by the bite of infected female phlebotomine sandflies, cause three main forms of leishmaniases: visceral (VL), cutaneous (CL), and mucocutaneous (MCL). There are an estimated 700,000 to 100,000,000 new cases of Leishmaniases annually in the world, widely distributed in tropical and subtropical climate zones, which lead to 26,000 to 65,000 deaths [1].

Chagas disease, also known as American trypanosomiasis, is found mainly in endemic areas of 21 continental Latin American countries, where it affects about 6 to 7 million people. Chagas disease is transmitted to humans by contact with feces or urine of triatomine bugs, known as “kissing bugs”. *Trypanosoma* cruzi infection is curable if treatment is initiated soon after infection; if the disease becomes chronic, the patient may develop cardiac, digestive, and/or neurological alterations [2].

Sleeping sickness is endemic in 36 sub-Saharan African countries, where is transmitted by tsetse flies. *Trypanosoma brucei gambiense* accounts for more than 98% of reported cases of the disease. Sustained control efforts have reduced the number of new cases so that in 2009 the number of reported cases dropped below 10,000 for the first time, and in 2018 there were only 977 cases recorded [3].

These diseases affect some of the poorest countries in the world and are often associated with malnutrition, population migration, poor housing, weak immune systems, such that they are generally recognized as neglected tropical diseases. However, the Mediterranean Basin is included in the affected areas, and climate change will exacerbate the ecological risk of human exposure in regions out of the current range of the disease; therefore, the issue concerns developed countries as well. Moreover, animal infection represents a further socio-economic problem: both domestic and wild animals are a reservoir for human infection, as in the case of endemic canine leishmaniasis in the Mediterranean area; in addition, livestock infection can cause significant economic losses in rural areas, as in the case of Nagana disease in Africa.

The therapeutic arsenal currently available for these diseases includes suramin, pentamidine, melarsoprol, and eflornithine for HAT; benznidazole and nifurtimox for Chagas disease; miltefosine, amphotericin B in liposomal formulation, pentavalent antimonials, and paromomycin for visceral leishmaniasis. Despite the need, these drugs are unsatisfactory because of a number of reasons: they are poorly effective, manifest severe side effects, episodes of resistance are increasingly frequent, and most treatments require prolonged and parenteral administration not suited for therapy in poor countries. Antimonials, for example, have a low therapeutic index and invoke extreme toxicities; therefore, they are administered only if strictly needed, in case of resistance to other treatments. Many different approaches have been attempted to date to develop new trypanocidal drugs, ranging from target based to phenotypic based and repositioning, and some compounds have been moved to clinical trials, but further efforts will be needed for new drugs to hit the market [4].

*Leishmania* and *Trypanosoma* parasites share many features, including gene conservation, high amino acid identity among proteins, the presence of subcellular structures such as glycosomes and the kinetoplastid, and genome architecture; such conservations may make drug development family-specific, rather than species-specific, i.e., based on the inhibition of a common, conserved target. Many unique metabolic pathways and cellular functions, divergent from other eukaryotes, are attractive target sources for drug discovery [4].

Oxidative stress plays an essential role in the host immune fight against infection, so that parasite survival mainly depends on the capability to resist this attack [5]. Infective trypanosomatids lack catalase [6] and other conventional redox controlling systems [7,8], and they base their defense on trypanothione, an unusual variant of glutathione, main actor in maintaining thiols’ homeostasis. Being essential and peculiar, trypanothione is a weakness for these parasites, and all related enzymes are considered interesting candidates for drug development [9]. Among these, trypanothione reductase (TR), the enzyme directly responsible for keeping trypanothione in the reduced state, has been extensively studied since it fulfills most of the requirements for a good drug target [10]. Indeed, TR is: (i) essential for parasite survival; (ii) absent in the host, in which TR is replaced by glutathione reductase (GR); (iii) druggable, in that it can be efficiently addressed by inhibitors.

TR has been validated as a target in both *Leishmania* and *Trypanosoma* as it is not possible to obtain TR-knockout mutants and its downregulation causes strong impairment of infectivity [11,12]. It has also been proven that antimonials, among the drugs currently in use to treat leishmaniasis, interfere with the trypanothione metabolism and inhibit TR [13,14], reinforcing the idea that targeting this protein is a concrete option for the treatment of these diseases. Moreover, the high sequence homology of TRs from different sources (80–100%) makes it a valuable target for developing a single, broad spectrum drug active against all trypanosomatids [15].

The main limitation of TR as a drug target lies in its high efficiency/turnover: it was shown that, in order to have a significant effect on parasite redox state and viability, TR activity must be reduced by at least 90%, meaning that only potent inhibitors, with submicromolar IC_50_, can be considered very promising lead compounds [11,12].

Many efforts have been made in order to find new effective hits through in vitro and in silico screening, in addition to the development of known scaffolds via SAR or structure-based design approaches [15,16,17,18,19,20,21,22,23,24,25,26,27,28,29,30,31,32,33], so that several classes of active compounds have been proposed to date.

This article provides an overview of the attempts made so far to rationalize the interaction between TR and known inhibitors through a detailed experimental structural characterization. Based on this information, as shown by some examples reported, it is possible to plan chemical modification of selected molecules in order to improve their potency and selectivity or to alter other characteristics, such as solubility, pharmacokinetics, and dynamics, without affecting affinity.

## 2. Relevant Structural Features of TR

Before addressing the binding mode of inhibitors, it is appropriate to describe the main structural features of TR, the mechanism of catalysis, and the recognition of substrates.

The structure of TR is thoroughly characterized, since the crystal structure has been solved for several species, namely *Crithidia fasciculata*, *L. infantum*, *T. brucei*, and *T. cruzi*, also in complex with substrates [14,34,35,36,37]. TR is an obligate homodimer with each of the two individual subunits, related by two-fold symmetry, comprising an FAD-binding domain (residues 1–160 and 289–360), an NADPH-binding domain (residues 161–288), and an interface domain (residues 361–488, *T. brucei* numbering).

The protein catalyzes the reduction of the dithiol trypanothione (from TS_2_ to T(SH)_2_) at the expense of the co-substrate NADPH. NADPH and TS_2_ bind different cavities facing opposite sides of the isoallosazine ring of FAD. The TS_2_ site, located at the interface between the two subunits, is shaped by residues belonging to both subunits. The reaction mechanism relies on the transfer of two electrons from NADPH to two catalytic cysteines (Cys52 and Cys57), via the FAD cofactor. Once the cysteines are reduced, the oxidized TS_2_ binds to the protein, and Cys52, deprotonated by the couple His461′-Glu466′, attacks the disulfide bridge of the substrate, resulting in the formation of a mixed disulfide. Finally, the attack of Cys57 on Cys52 enables the release of the reduced T(SH)_2_ (Figure 1). During catalysis, no major structural changes occur, apart from the strictly necessary displacements of the side chains of the residues involved.

The structure is almost identical for all the characterized species, in accordance with the high degree of sequence similarity (Figure 1, panel D). Indeed, TRs from all Trypanosomatidae share at least 67% of primary sequence, with >82% identity among *Leishmania* spp. and >80% among *Trypanosoma* spp. Similarity reaches 100% for residues shaping both substrates’ binding sites, with the fact that the mode of binding of ligands is the same for all TRs characterized to date [34,35,36]. The trypanothione assumes variable conformations in the wide cavity, as an effect of the “dynamics” of its binding. In fact, trypanothione enters as a disulfide but, upon reduction, it is released in an extended conformation. Despite this variability, some interactions emerge to be particularly relevant and specific for binding: Glu18, together with other acidic residues, accounts for the positive charge of the substrate, while the almost hydrophobic patch including Trp21, Tyr110, and Met113 mediates the interaction with the polyamine moiety contained in trypanothione.

This observation suggests that, in the search for new inhibitors, results can be transferred from one TR to the others, and the chance exists to find a common inhibitor active on all TRs that can lead to the development of a broad spectrum trypanocidal drug. This is clearly an ambitious goal because differences in biology and lifestyle of these parasites, although they are closely related genetically, could cause species-specific efficacy, as in the case of eflornithine (DFMO). Indeed, DFMO, a suicide inhibitor of ornithine decarboxylase from any source, is the treatment of choice for advanced stage of the sleeping sickness caused by *T. brucei* but is rather ineffective on other infections [9].

## 3. Off-Target Evaluation: Comparison with Glutathione Reductase (GR)

Selectivity is a fundamental parameter in the evaluation of a potential pharmacological target. For the development of an antiparasitic drug, it is important to choose a target that has substantial differences compared to the host homolog(s), the so-called off-target(s), in order to promote specific action and minimize side effects.

The trypanothione/TR couple replaces many of the antioxidant and metabolic functions of the glutathione/glutathione reductase (GSH/GR) and thioredoxin/thioredoxin reductase (Trx/TrxT) systems present in the host [38].

GR is the closest human homolog of TR as they have the same overall fold, with 38% sequence identity, and catalyze the same reaction on very similar substrates. Both GR and TR reduce a disulfide bridge that is intermolecular for GR (GSSG→2 GSH) and intramolecular for TR (TS_2_→T(SH)_2_). Indeed, trypanothione is an analog of glutathione, comprised of two glutathione molecules linked by amide bonds occurring between the glycyl carboxylate groups of each GSH and the primary amines of the polyamine spermidine.

The most significant differences between the two proteins reflect the differences between their cognate substrates: TS_2_ is bulkier than GSSG and positively charged due to the spermidine moiety, while GSH has a net negative charge at physiological pH. As a consequence, the TS_2_ binding site in TR is wider and negatively charged with respect to the GSSG binding site in GR (Figure 2). In particular, selective interactions take place between the spermidine moiety and residues Glu18, Trp21, Ser109, Tyr110, and Met114 that are not conserved in GR and are partially replaced by arginine residues (Arg37, Arg38, and Arg347).

These steric and electrostatic differences account for the selectivity for substrates [39] and emphasize the potential to generate parasite-specific compounds.

## 4. Structural Characterization of TR Inhibitors

Structural studies on TR, intensified over the past 10 years, strongly improved the understanding of the molecular basis of ligand binding, allowing to identify hot-spots for interaction with substrates and inhibitors. This knowledge has been exploited in few structure-based design approaches which, in some cases, have led to a significant improvement in the performances of lead molecules [27,37,40].

To date, the crystallographic structure of TR in complex with 21 different inhibitors has been defined (see Table 1 and Appendix A and Figure 3). These can be grouped into 3 main inhibition modes: (i) competition with trypanothione, due to binding to the wide TS_2_ cavity, comprising most of the characterized inhibitors; (ii) competition with NADPH, due to the binding to NADPH cavity; (iii) redox cysteines inactivation, due to a metal binding to Cys52 and Cys57 in the catalytic site. A fourth inhibition mode has been recently proposed [41], based on the disassembly of the TR dimer induced by small molecules designed to interfere with protein–protein interaction. However, poor structural information is available for this case.

Below is a description of the various classes of characterized inhibitors, divided according to mode of action and binding site, with attention to how the structural information was used to guide the design of better molecules.

### 4.1. Inhibitors Targeting the TS_2_ Binding Cavity

As mentioned above, TR has a wide active site, suited to accommodate the voluminous trypanothione substrate. Most of the characterized inhibitors bind to this cavity, mainly in the so-called “mepacrine binding site” (MBS), a hydrophobic patch located at the entrance. Fewer ligands bind deeper in the cavity, closer to the real catalytic site, where the redox cysteines are located and TS_2_ reduction takes place (Figure 4).

#### 4.1.1. Mepacrine Binding Site (MBS)

Mepacrine, also named quinacrine, is a well-known antiprotozoal compound, superseded by safer and more effective agents. In 1996, Jacoby and coworkers described the crystal structure of *T. cruzi* TR in complex with mepacrine [42] (coordinates not available in the PDB). The ligand, known to compete with TS_2_, was found on the edge of the active site. Later, Saravanamuthu and coworkers [43], added details to this interaction, by solving the structure of TR with an alkylating mepacrine derivative at higher resolution. The interaction is dominated by 4 residues, namely Trp21, Met113, Tyr110, and Glu18. They found that two molecules of the inhibitor bind in a synergistic way by stacking of planar acridine ring, thereby gaining an increased number of binding interactions. In particular, the aromatic acridine ring of the first mepacrine molecule stacks over Trp21 and is lined by Met113, while the alkylamino chain points inside the active site, held in position by Glu18, and covalently binds to Cys52 (*T. brucei* numbering); the second stacked acridine is lined by Tyr110. The site immediately turned out to be interesting since the residues that shape it are important for TS_2_ binding and are not conserved in GR. Indeed, mepacrine does not affect human GR.

Since then, other scaffolds besides tricyclic acridine have been found to bind to MBS. In 2011, Patterson et al. [37] developed a new class of TR inhibitors based on a 3,4-dihydroquinazoline scaffold, by an elegant combination of chemical-driven and structure-based approaches. Starting from a high-throughput screen hit [17], indicated as compound 1a, they composed and tested a small commercial collection of molecules. On the basis of the structure–activity relationship (SAR) analysis of these compounds, the authors planned the synthesis of new derivatives. The crystal structure of hit 1a (WP5 in PDB) and another 3 representative inhibitors in complex with TR from *T. brucei* revealed the mode of binding and helped to rationalize SAR analysis. All derivatives bind to the TS_2_ cavity at the MBS and surprisingly induce a structural variation of the active site that was revealed to be critical for binding. Indeed, a new subpocket, which accommodates the C4-phenyl substituent of the scaffold, is generated by the displacement of Met113 side chain. Structural information was subsequently used to design other inhibitors, including analogs that challenged the induced subpocket. Overall, this hit-to-lead approach resulted in the development of inhibitors with improved potency, among which the best performing has a 30-fold lower IC50 for *T. brucei* TR with respect to starting compound (1a, WP5 in PDB: 6.8 μM; 29a, WPF in PDB: 0.23 μM), although selectivity remained an issue (Figure 5).

In 2013, Ilari and collaborators [35] described the binding of a diarylpyrrole to TR from *L. infantum*. The compound was selected from an in-house collection on the base of activity on amastigote form of *L. donovani* and docking studies on TR. The structure shows that, as observed for mepacrine derivative, two molecules bind to MBS without inducing any variation in the cavity, but the mode of binding differs considerably. In this case, the compound assumes multiple conformations, likely due to its intrinsic flexibility, and no stacking to Trp21 takes place, indicating that MBS has the capability to interact in different ways with unrelated scaffolds.

Derivatives of 1-(1-(Benzo[b]thiophen-2-yl)cyclohexyl)piperidine (BTCP), another class of compounds able to bind MBS, are probably, to date, the most explored compounds for structure-based development. Identified by HTS together with many other tricycles [21], the lead BTCP was found to be a competitive inhibitor of TbTR, active on *T. brucei* cultures but endowed with poor selectivity against mammalian cells. However, it was considered to be a promising screening hit for further development due to some drug-like characteristics such as low molecular weight, lack of activity on GR, capability of crossing the blood–brain barrier, and critical property for treating HAT [44]. The first attempt to describe the binding to plan a structure-based improvement of BTCP’s properties was carried out by Persch et al. [40]. Previous work suggested that binding of BTCP occurs at the so-called Z-site, a hydrophobic region in front of MBS [45]. However, the conjunction of mutation studies and virtual ligand docking simulations led to the prediction that the binding takes place at MBS. This was confirmed by the co-crystal structure of both *T. brucei* and *T. cruzi* TR with compound 10a (2JR 3-digit code in PDB), a BTCP analog in which a thiazole is inserted between indole and cyclohexyl rings (Figure 6). Two key interactions appear to control binding: the protonated tertiary amine of the ligand makes a Coulombic interaction with Glu18, while the indole moiety binds to the hydrophobic wall of MBS (Trp21, Tyr110, Met113) even if it adopts different orientations in the two structures [40].

Further efforts for improving properties and potency of this class have been recently undertaken by De Gasparo and colleagues [27,46]. They explored the possibility to combine the 2 different binding modes observed for compound 10a by introducing other substituents on the thiazole to be able to increase water solubility and binding affinity. At first, three new series of BTCP derivatives were synthesized and tested for activity on TR and parasites, but the results were not very satisfactory in terms of efficacy, though useful structural information emerged [46]. In fact, co-crystal structures of 2 new ligands (18 and 19) confirmed the mode of binding previously observed, with the indolyl-thiazole core adopting identical orientation, and the newly introduced water-solubility-providing substituents oriented toward the periphery of the active site.

Later, new substitutions resulted in a significant improvement of potency and selectivity. Indeed, compound (+)-2 (M9J in PDB), claimed to be the most effective non-covalent inhibitor of TR ever reported, inhibits *T. brucei* TR with an inhibition constant K_i_ of 73 nM and is fully ineffective against human GR, even if its toxicity against mammalian cells is relatively high [27]. Two major structural changes led to this result: the modification of the substituent on the indole moiety, combined with the introduction onto position 4 of the central thiazole moiety of a propargylic substituent, designed to target a hydrophobic sub-pocket near the catalytic cysteines in the TR active site. The structure of inhibitor (+)-2-TR complex confirmed the prediction, showing that the indole protonated substituent expands the interaction in MBS to Asp116, while the propargylic moiety, although mobile, locates deeper into the cavity. Moreover, a HEPES molecule, found in close proximity to propargylic substituent, suggests the opportunity to further modify the lead to reach another anchor point in the wide TS_2_ cavity (Figure 6).

Very recently, a new spiro-containing series has been found to bind trypanothione cavity, resembling the mode of binding of compound M9J [47]. The hit, identified by HTS on *T. brucei* TR, was found to inhibit both the recombinant enzyme and the enzyme in cell lysate, as well as parasite proliferation in the low micromolar range (2–5 µM) while being inactive on human GR. Crystallographic studies confirmed the hot-spots for interaction already found for BTCP-derivatives (Figure 7). Indeed, the phenyl-triazaspiro core anchors the molecule to the MBS (Trp21, Met113, and Tyr110) while the arms cause steric hindrance both at the bottom and at the entrance of the cavity: the tertiary amino group of the hydrophilic carboximidamide arm, fluctuating toward the entrance, engages a weak electrostatic interaction with Glu18 and, in general, with the negative environment; the hydrophobic bicycle-heptane moiety extends deeper in the cavity, pointing to the same hydrophobic sub-pocket targeted by the propargylic substituent of (+)-2 BTCP-derivative (Val53, Val58, Ile106, and Leu399). A second binding site is located at the dimeric interface but seems to be not significant for activity. Though several rounds of optimization are needed, what makes this spiro-core particularly interesting is the fact that, as for BTCP, molecules containing this moiety are known to be able to penetrate the brain, a very appealing characteristic for the treatment of the second stage of sleeping sickness, affecting the central nervous system [48,49].

#### 4.1.2. Catalytic Site

Other compounds have shown the capability to bind the inner part of the active site. Recently, screening of an in-house collection detected a novel class of diarylsulfides active on *Leishmania* culture and TR [32]. In particular, the compound RDS777 (6-(sec-butoxy)-2-((3-chlorophenyl)thio)pyrimidin-4-amine) was found to inhibit TR with high efficiency (K_i_ 0.25 μM) by competing with TS_2_, and to affect parasites in the micromolar range (IC50 29 μM). The crystal structure of RDS777 (RDS in PDB) in complex with *L. infantum* TR revealed the binding of 4 inhibitor molecules, one of which lays at the bottom of TS_2_ cavity, in direct contact with catalytic site by establishing hydrogen bonds with the residues involved in catalysis, namely Glu466’, Cys57, and Cys52. A second molecule is found in one out of two cavities, placed closer to the entrance, engaged in a stacking interaction with the first one. The other two molecules interact with the NADPH-binding site and are discussed later. Based on the structural information, a series of new derivatives have been synthesized, one of which has a higher activity on parasite cultures (IC50 11 μM) and is able to decrease the reduced-T(SH)_2_ concentration in cell [25]. However, this new compound is less effective in TR inhibition (K_i_ 12 μM) and docking studies suggest that it prefers the second outermost binding site, indicating that it likely has other intracellular targets besides TR.

Other diaryl sulfides have been proposed previously and the binding site, predicted using docking, was different from RDS777, corresponding to the MBP and Z-site [50,51]. However, it must be considered that the bond is plausibly influenced more by the nature of the aryl substituents than by the thioether itself.

#### 4.1.3. Metal Inhibitors

Metalloid-based drugs, such as pentavalent antimonials and arsenicals, are currently used to treat trypanosomiasis and leishmaniasis, despite having severe side effects and resistance phenomena [52]. It is known that these drugs, at least in part, act on TR by binding catalytic cysteines. In particular, Baiocco et al. [14] demonstrated that Sb(III) efficiently inhibits reduced TR (K_i_ 1.5 μM) by forming a stable complex with the residues involved in catalysis, namely the two cysteines (Cys52 and Cys57), His461 (the residue that together with Glu466′ activates the Cys52 similar to the cysteine proteases), and Thr335. Besides antimony, silver and gold were proven to bind TR in a similar way but even more efficiently [53,54,55] with K_i_ down to 20 nM. Particularly interesting is the case of auranofin, a gold-containing drug used to treat rheumatoid arthritis [56]. Tested on *Leishmania* TR and parasites [54], auranofin was found to be 10-fold more potent than Sb on TR (K_i_ 0.15 μM) and, most crucially, it acts via a double mode of action. In fact, besides expected gold complexation, the thiosugar moiety of auranofin contributes to inhibition by binding the inner part of TS_2_ site. This finding suggests the opportunity to combine scaffolds that are able to bind the outer TS_2_ cavity with auranofin or other metal-coordinating moieties to exploit double inhibition mechanism and to promote a selective targeting of otherwise poorly specific metal inhibitors (Figure 8).

### 4.2. Inhibitors Targeting NADPH Binding Cavity

NADPH-binding cavity is considered less appealing for the development of specific TR inhibitors clearly due to the nature of this ubiquitous cofactor involved in a number of pathways in all organisms. Nevertheless, a couple of TR inhibitors have been found to bind to this site, one of which deserves some attention.

As anticipated, diaryl sulfide RDS777 was found to bind to even the NADPH-binding site, specifically at the entrance where adenosine moiety of NADPH usually binds [32], though kinetic characterization denied competition for the cofactor so it can be speculated that binding is weak or due to crystallographic artifact.

In 2018, a new inhibitor targeting the NADPH-binding site was identified by HTS on *L. infantum* TR, based on a new luminescent assay, followed by extensive SAR evaluation [23]. The inhibitor is not particularly potent (IC50 for TR 7.5 μM) but it is interesting due to some other characteristics. Indeed, it competes for NADPH but is inactive on human GR and thioredoxin reductase, and it has dose-dependent anti-proliferative effect on *L. infantum* promastigotes at micromolar concentrations (IC50 12.4 μM). Crystallographic analysis of the complex revealed that the compound binds at the entrance of NADPH site, similar to RDS777, in a pocket not conserved in human GR (Figure 9). Even if cytotoxicity data are not available and the compound could be active on other NADPH-dependent human enzymes, it represents the first proof of the existence of a druggable site in NADPH cavity.

### 4.3. Nonpeptidic Dimerization Inhibition

The disruption of the functional dimer of TR by targeting PPI has been recently proposed as an intriguing strategy alternative to competitive inhibition. Starting from the analysis of the dimerization interface of *L. infantum* TR, Toro and colleagues identified few interaction hot-spots involving an α-helical element from which they derived linear and cyclic peptides that are able to strongly affect both dimerization (95% decrease) and activity in the low micromolar range as well as *Leishmania* viability in vitro [57]. Later, in order to improve drug-like properties such as stability and permeability, nonpeptidic small molecule analogs were synthesized and tested with modest results, in which they showed a drop in efficacy for both dimerization and activity. Attempts to gain structural information on the interaction with best performing peptidomimetics (IC50: 5–9 μM) failed due to protein precipitation, possibly induced by dimer disruption (~30% at 20 μM concentration). Instead, a mild inhibitor (IC50: 52.2 μM), inactive on dimerization, was unexpectedly found to bind the MBS. Interaction involves stacking of pyrrolopyrimidine core on Trp21 and H-bond interactions of amide groups with Glu18 and Ser109, while the rest of the molecule protrudes out of the cavity [41].

## 5. Conclusions and Perspectives

Among the many pathways proposed as potential targets for antitrypanosomatid drug development, trypanothione metabolism is one of the most explored due to its critical role in redox homeostasis and its peculiarity. TR has been considered a promising target since its discovery because it satisfies most requirements for candidates, being essential, unique and druggable. Various inhibitor series have been identified and proposed as lead compounds but, to our knowledge, none of them has been advanced to clinical trials. Common reasons for that are sub-optimal potency, poor selectivity leading to toxicity, low bioavailability or biodistribution causing inactivity on animal models.

In this context, structural characterization of inhibitor binding offers precious aid to improve inhibitor performances through structure-based design. In the last few years, several studies explored TR-inhibitor interaction by X-ray crystallography, revealing important information for binding rationalization and future development. Indeed, most inhibitors characterized so far locate to the wide trypanothione cavity, mainly at the entrance of the so-called mepacrine binding site (MBS). The MBS resulted to be quite promiscuous; in fact, besides the polyamine moiety of the substrate, it is able to bind different aromatic scaffolds. Binding site promiscuity can be an advantage in drug development because it favors polypharmacology approaches, known to decrease emergence of resistance. Moreover, it allows researchers to select molecules with convenient characteristics, as is the case for two identified scaffolds, BTCP and spiro-moiety, that are able to cross the BBB, a desirable feature for the treatment of HAT.

Besides the MBS, other hot-spots have been identified in the trypanothione cavity. A hydrophobic subpocket, located deeper in the site, accommodates the hydrophobic arm of two different inhibitors, one of which displays nanomolar activity on TR. Finally, metal ions such as Sb(III), Ag(I), and Au(I) have proven to target redox-active cysteines in the catalytic site, confirming one of the proposed mechanisms of action for antileishmanial antimonial therapy [13,14].

The broadness of the trypanotione cavity is believed to be responsible for the relatively low potency of most inhibitors identified to date, that show inhibition constants in the low micromolar range, not enough for effective action. However, the identification of multiple hot-spots for interaction provides the chance to merge different scaffolds in one molecule, as in the serendipitous case of auranofin, in order to increase efficacy and selectivity, keeping in mind the size limitations for drug-like compounds.

Despite of the most rational approach, recent results have shown that the search for lead compounds should not focus solely on the trypanothione cavity: the identification of an inhibitor addressing the NADPH cavity, selective against the main off-target GR, as well as peptides and peptidomimetics interfering with TR dimerization give a proof of concept for the idea that other sites can be exploited for TR inactivation.

It cannot be excluded that unexpected effective binding sites exist. Given the availability of well diffracting crystals for both *T. cruzi* and *T. brucei* TR, a fragment-based screen campaign could reveal new small organic molecules suitable as lead compounds, targeting already known sites or unexplored hot-spots for new mechanisms of inhibition.

## Figures and Tables

**Figure 1 molecules-25-01924-f001:**
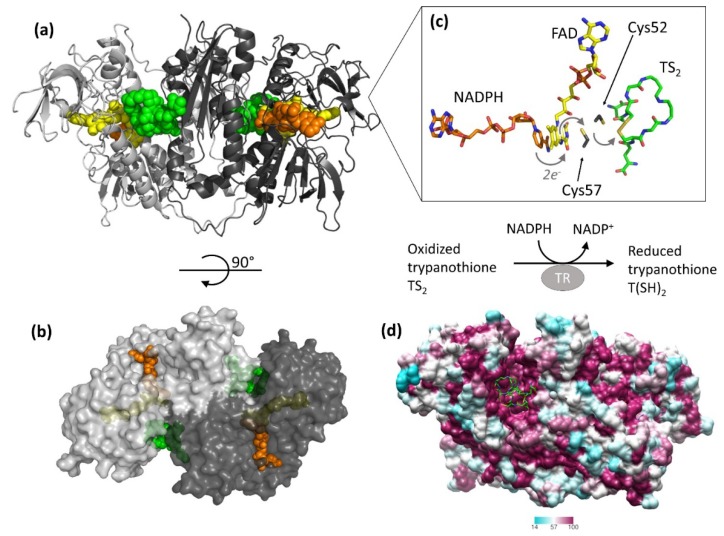
Structure and activity of trypanothione reductase (TR). (**a**,**b**) Two views of TR dimer from *T. brucei* (PDB: 2wow) are shown. NADPH (orange), FAD (yellow), and trypanothione (green) are represented as spheres to highlight the binding sites. (**c**) The detail shows all entities involved in the electron transfer from NADPH to trypanothione. For clarity, trypanothione is depicted as modeled in TR from *T. cruzi* (PDB: 1bzl), where a single oxidized conformation is observed in the absence of NADPH. (**d**) Sequence conservation of TR. The dimer of TR from *T. brucei* (PDB: 2wow) is colored according to the percentage of amino acid identity with respect to other representative TR sequences (*C. fasciculata*, *T. cruzi*, *T. congolense*, *T. brucei*, *L. braziliensis*, *L. infantum*, *L. major*). Trypanothione, represented as green sticks, assumes multiple conformations in the wide and highly conserved binding cavity.

**Figure 2 molecules-25-01924-f002:**
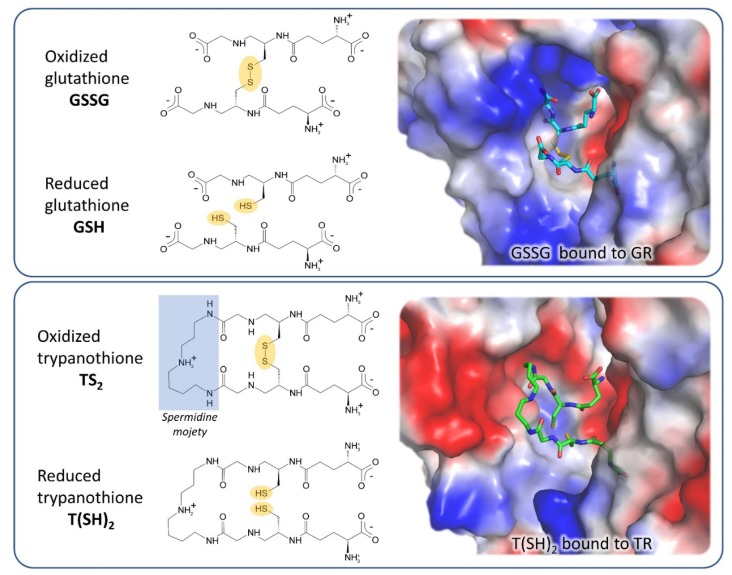
Substrates and active site of glutathione reductase (GR) and TR. The comparison between the electrostatic potential surfaces of GR (upper panel, PDB: 1gra) and TR (lower panel, PDB: 1bzl) highlights the difference in size and charge of substrate binding sites, related to substrates features.

**Figure 3 molecules-25-01924-f003:**
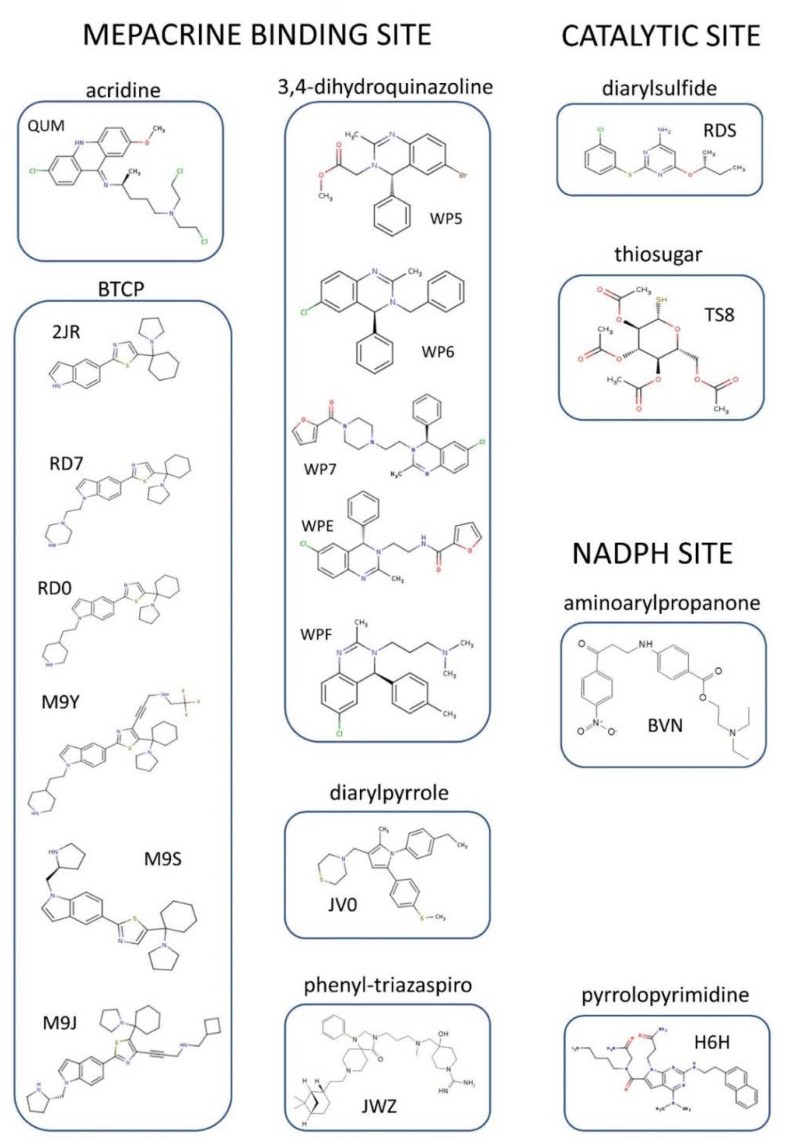
Organic inhibitors co-crystallized with TR, grouped by binding site and molecular scaffold. Molecules are named by PDB 3-digit ID.

**Figure 4 molecules-25-01924-f004:**
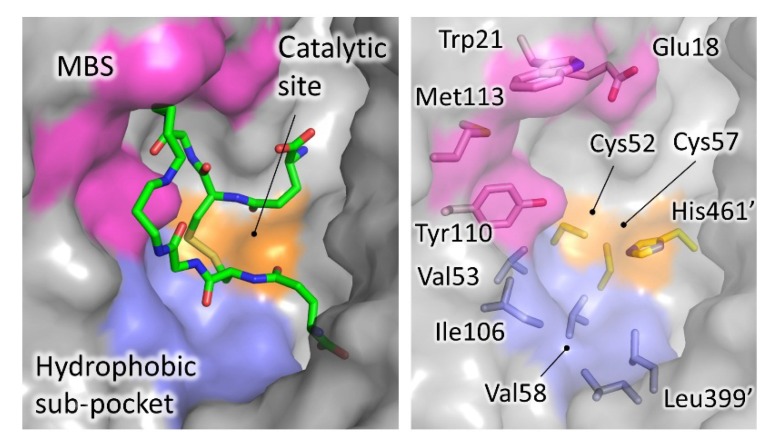
Trypanothione binding cavity. Surface representation of TR from *T. brucei* (pdb: 2wow). The most significant areas for ligand interaction are highlighted with different colors. TS_2_, extracted from *T. cruzi* structure (pdb: 1bzl) is shown as sticks.

**Figure 5 molecules-25-01924-f005:**
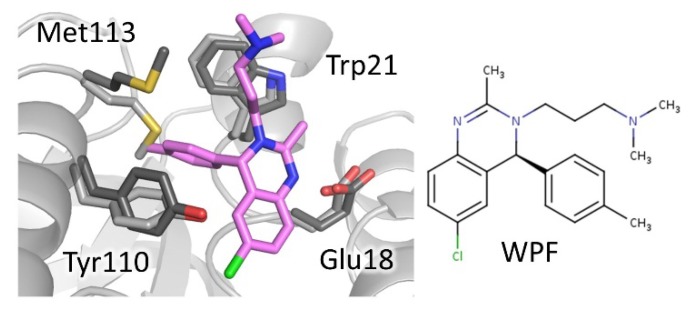
Mode of binding of 3,4-dihydroquinazoline derivatives. Compound WPF, best performing compound of the series, binds to mepacrine binding site (MBS). Critical residues are shown as sticks, light grey for inhibitor-free (PDB: 2wow) and dark grey for inhibitor-bound TR (PDB: 2wpF). Note the displacement of Met113 to accommodate the C4-phenyl substituent.

**Figure 6 molecules-25-01924-f006:**
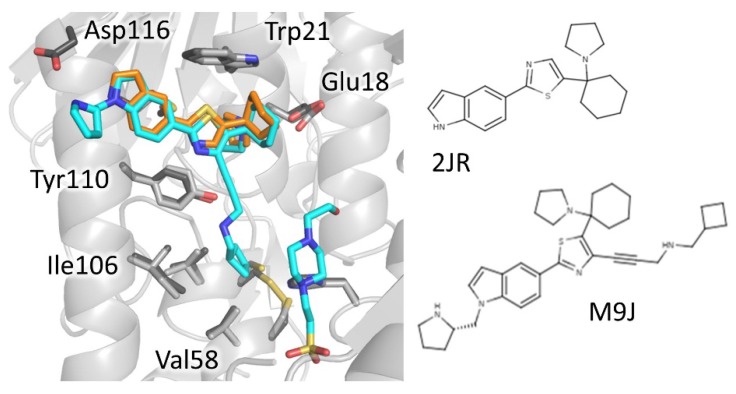
Mode of binding of 1-(1-(benzo[b]thiophen-2-yl)cyclohexyl)piperidine (BTCP)-derivatives. Compound 2JR (orange) binds to MBS while its best performing evolution M9J (cyan) extends to the catalytic site and, in addition, a HEPES molecule binds deeper in the cavity (cyan). Critical residues are shown as sticks, light grey for inhibitor-free (PDB: 2wow) and dark grey inhibitor-bound TR (PDB: 6oez).

**Figure 7 molecules-25-01924-f007:**
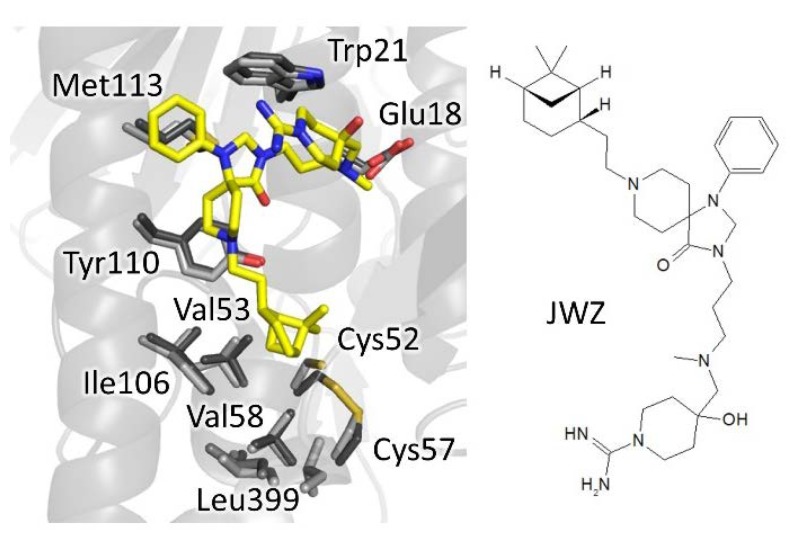
The binding mode of spiro-containing derivative. The phenyl-triazaspiro core seizes the MBS while the bicyclo-heptane moiety accomodates in the hydrophobic sub-pocket. Involved residues are shown as sticks, light grey for inhibitor-free (PDB: 2wow) and dark grey for inhibitor-bound TR (PDB: 6rb5).

**Figure 8 molecules-25-01924-f008:**
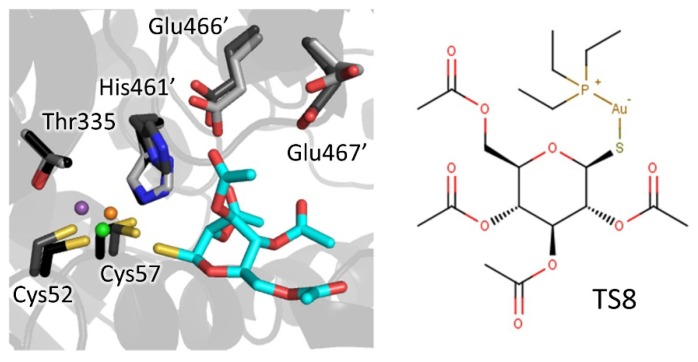
Metal complexation: Sb(III) (violet) forms a stable complex with catalytic residues. Similarly, Au(I) (orange) displays a planar-trigonal coordination (Ilari et al., 2012) with Cys52, Cys57, and a chloride ion (green). The thiosugar moiety (cyan) participates in the inhibition mechanism interacting with His466′, Glu466′, and Glu467′. Residues are shown as sticks, light grey for inhibitor-free TR (PDB: 2jk6), dark grey for Au-bound TR (PDB: 2yau), and black for Sb-bound TR (PDB: 2w0h).

**Figure 9 molecules-25-01924-f009:**
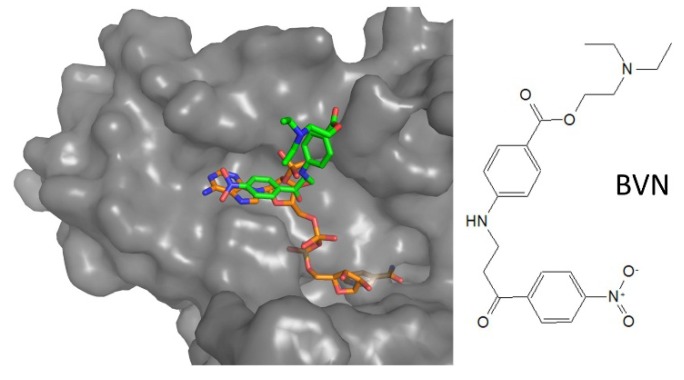
Mode of binding of aminoarylpropanone-derivative. Compound BVN (green) binds to NADPH (orange) binding site, where adenosine moiety locates. Protein surface is shown for BVN-bound LiTR (PDB: 6er5), superposed to NADPH-bound structure (PDB: 2w0h).

**Table 1 molecules-25-01924-t001:** All inhibitors co-crystallized with TR.

Site	Scaffold	PDB Code	Source	Inhibitor PDB ID (Paper ID ^a^)	Potency ^b^	Reference
**MBS**	Acridine 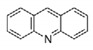	*Not available*	Tc	(Quinacrine or mepacrine)	K_i_: 25 μM	Jacoby, 1996
1gxf	Tb	QUM(Quin. mustard)	*Irreversible inhibition*	Saravanamuthu, 2004
3,4-dihydroquinazoline 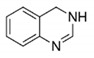	2wp5	Tb	WP5(1a)	IC_50_: 6.8 μM	Patterson, 2011
2wp6	Tb	WP6(6a)	IC_50_: 0.93 μM
2wpc	Tb	WP7(13e)	IC_50_: 0.42 μM
2wpe	Tb	WPE(11e)	IC_50_: 0.86 μM
2wpf	Tb	WPF(29a)	IC_50_: 0.23 μM
BTCP 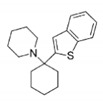	4nev	Tb	2JR(10a)	Ki: 12 μMInh. [%] ^b^: 43	Persch, 2014
4new	Tc	2JR(10a)	Ki: 4 μMInh. [%] ^b^: 79
6btl	Tb	RD7(18)	K_i_: 3.8 μMInh. [%] ^c^: 80	De Gasparo, 2018
6bu7	Tb	RD0(19)	K_i_: 6.4 μMInh. [%] ^c^: 78
6oez	Tb	M9J((+)-2)	K_i_: 73 nM	De Gasparo, 2019
6oey	Tb	M9S((+)-4))	K_i_: 2.1 μM
6oex	Tb	M9Y(5)	K_i_: 1.5 μM
diarylpyrrole	4apn (B)	Li	JV0(1)	K_i_: 4.6 μMIC_50_: 13.8 μM	Baiocco, 2013
phenyl-triazaspiro	6br5	Tb	JWZ(1)	IC_50_: 5.7 µM	Turcano, 2020
Pyrrolopyrimi- dine	6i7n(B)	Li	H6H(2f)	IC_50_: 52.2 µM	Revuelto, 2019
**Catalytic site**	diaryl sulfide	5ebk	Li	RDS(RDS 777)	K_i_: 0.25 µM	Saccoliti, 2017
**Catalytic site/cysteines**	Metal/thiosugar	2yau	Li	AU-TS8(auranofin)	K_i_: 0.15 µM	Ilari, 2012
**Catalytic cysteines**	Metal	2w0h	Li	SB	K_i_: 1.5 µM	Baiocco, 2009
2x50	Li	AG	K_i_ (Ag1): 500 nMK_i_ (Ag0): 50 nM	Baiocco, 2010
**NADPH-cavity**	3-amino-1-arylpropan-1-one	6er5	Li	BVN(3)	IC_50_: 12.4 µM	Turcano, 2018

^a^ identification code of the inhibitor as reported in the original paper. ^b^ K_i_, IC_50_ and/or percentage of inhibition are reported when available in literature. ^c^ Percent inhibition by 40 mM inhibitor in the presence of 40 mM dithiol trypanothione (TS_2_).

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
