# Peer review of "Targeting Trypanothione Reductase, a Key Enzyme in the Redox Trypanosomatid Metabolism, to Develop New Drugs against Leishmaniasis and Trypanosomiases"

_molecules, 2020, doi:10.3390/molecules25081924_

Round 1
Reviewer 1 Report
this review describes drug discovery efforts to target the important drug target Trypanothione reductase in various human pathogenic parasites. this is a very detailed and well researched review which i am happy to suggest is published after some minor adjustments (mainly to the written English) are made which are detailed in the PDF which i have annotated and is attached to this review.

Author Response
We thank the reviewer for appreciating the manuscript and for the useful suggestions.
We corrected all the linguistic inaccuracies and inserted the required references, apart for line 260 since we could not find any indication for even hypothetical off-targets for BTCP in mammals.
Rephrased sentences have been highlighted in red in the new version of the manuscript.
Reviewer 2 Report
The manuscript "Targeting trypanothione reductase, a key enzyme in the redox trypanosomatid metabolism, to develop new drugs against Leishmaniasis and Trypanosomiases" is a very useful review and very well written. I would recommend acceptance of the manuscript with minor revision. I have a few minor suggestions which might improve the manuscript.
(1) Page 2 line 57 the authors may want to state the current drugs used to treat the disease.
(2) page 2 line 82, here it might be a good idea to state that antimonials have a low therapeutic index and invokes extreme toxicities and therefore best to avoid if possible.
(3) page 8 line213 clarify if it is Jacoby or Jacobi (be consistent with the table on page 7).
(4) page 13 line 414 please consider changing to "change to is quite promiscuous."
(5) page 13 line 415-419 it is better to state here that this might also invokes side effects
(6) The reference has to be consistent. Please try and follow the format for all the references: author names, title, year, volume (issue), page numbers (first and last). Many of the references do not follow this pattern.
Author Response
We thank the reviewer for appreciating the manuscript and for the useful suggestions.
Find below point-by-point response to the reviewer’s comments.
(1) Page 2 line 57 the authors may want to state the current drugs used to treat the disease.
Answer: we inserted the following sentence:
The therapeutic arsenal currently available for these diseases includes: suramin, pentamidine, melarsoprol and eflornithine for HAT; benznidazole and nifurtimox for Chagas disease; miltefosine, amphotericin B in liposomal formulation, pentavalent antimonials and paromomycin for visceral leishmaniasis.
(2) page 2 line 82, here it might be a good idea to state that antimonials have a low therapeutic index and invokes extreme toxicities and therefore best to avoid if possible.
Answer: we inserted the following sentence in the previous paragraph:
Antimonials, for example, have a low therapeutic index and invokes extreme toxicity, therefore they are administered only if strictly needed, in case of resistance to other treatments.
(3) page 8 line213 clarify if it is Jacoby or Jacobi (be consistent with the table on page 7).
Answer: Jacoby, corrected in table.
(4) page 13 line 414 please consider changing to “is quite promiscuous."
Answer: we think that the current form “resulted to be quite promiscuous” is appropriate.
(5) page 13 line 415-419 it is better to state here that this might also invokes side effects
Answer: Here the term promiscuity refers to the binding site that is capable to bind different molecular scaffolds, it does not refer to a drug able to bind different targets. In the case of a target for antiparasitic drugs, we cannot imagine any source of side effects for a promiscuous binding site: any compound able to bind the target is welcome.
(6) The reference has to be consistent. Please try and follow the format for all the references: author names, title, year, volume (issue), page numbers (first and last). Many of the references do not follow this pattern.
Answer: We modified the references according to your suggestion.